# Facile Bioinspired Preparation of Fluorinase@Fluoridated Hydroxyapatite Nanoflowers for the Biosynthesis of 5'-Fluorodeoxy Adenosine

**Ningning Li [1], Bingjing Hu [1], Anming Wang [1,\*], Huimin Li [1], Youcheng Yin [2], Tianyu Mao [1] and Tian Xie [2,\*]**

1   College of Materials, Chemistry and Chemical Engineering, Hangzhou Normal University, Hangzhou 311121, China; liningning1201@gmail.com (N.L.); binjinghu@gmail.com (B.H.); huiminli0919@gmail.com (H.L.); tianyumao11@gmail.com (T.M.)
2   Holistic Integrative Pharmacy Institutes, College of Medicine, Hangzhou Normal University, Hangzhou 311121, China; youchengyin@gmail.com
\*   Correspondence: waming@hznu.edu.cn (A.W.); xbs@hznu.edu.cn (T.X.); Tel.: +86-571-2886-5978 (A.W.)

**Abstract:** To develop an environmentally friendly biocatalyst for the efficient synthesis of organofluorine compounds, we prepared the enzyme@fluoridated hydroxyapatite nanoflowers (FHAp-NFs) using fluorinase expressed in *Escherichia coli* Rosetta (DE3) as the biomineralization framework. The obtained fluorinase@FHAp-NFs were characterized by scanning electron microscope (SEM), X-ray diffraction (XRD), and FT-IR spectrum and used in the enzymatic synthesis of 5'-fluorodeoxy adenosin with S-adenosyl-L-methionine and fluoride as substrate. At an optimum pH of 7.5, fluorinase confined in the hybrid nanoflowers presents an approximately 2-fold higher synthetic activity than free fluorinase. Additionally, after heating at 30 °C for 8 h, the FHAp-NFs retained approximately 80.0% of the initial activity. However, free enzyme could remain only 48.2% of its initial activity. The results indicate that the fluoride and hybrid nanoflowers efficiently enhance the catalytic activity and thermal stability of fluorinase in the synthesis of 5'-fluorodeoxy adenosine, which gives a green method for producing the fluorinated organic compounds.

**Keywords:** organofluorine compound; fluorinase; 5'-fluorodeoxy adenosine; biomineralization; immobilization; fluoridated hydroxyapatite nanoflowers

## 1. Introduction

For pharmaceuticals, fluorination can affect the binding and affinity of a candidate drug by adjusting the acidity and basicity of the compound [1]. Therefore, approximately only 2% of fluorinated drugs were on the market in 1970, and the current number has grown to approximately 25% [2]. To date, most organofluorine compounds were synthesized using chemical methods [3–5], and the biosynthesis of organofluorine compounds has only been used to investigate the catalysis of the corresponding fluorinase [6–8]. In chemical methods, fluorination is mainly carried out in nucleophilic, electrophilic, radical, and transition metal-catalyzed fluoroalkylation reactions [9]. A feature of nucleophilic fluoroalkylation is that the fluoroalkyl group is transferred to an electrophilic group, which is equivalent to a fluorocarbon anion or a fluoroalkyl metal species [9,10]. Through an electrophilic reaction [11], the promotion of lithium hexamethyldisilazide (LiHMDS) could be utilized to achieve the direct $\alpha$-difluoromethylation of carbonyl compounds such as esters and $CF_3H$. Fluoroalkyl radicals also play an important role as reactive intermediates in organofluorine chemistry [12], and AgF can act as a stabilizing intermediate to mediate fluorinative cross-coupling to prepare $\alpha$-$CF_3$ alkenes and $\beta$-$CF_3$ ketones conveniently. However, for the chemical synthesis of some fluorinated

pharmaceuticals such as 5'-fluorodeoxy adenosine, the process is cumbersome, and the final synthesis yield is low—approximately only 20% [13]. In addition, for the fluorine substitution reagents used therein, including trifluoromethane and trifluoroacetic acid, trifluoromethane is heated and explains the highly toxic fumes, while trifluoroacetic acid is highly corrosive and irritating, causing burns to the human body.

Compared with chemical methods, biocatalytic synthesis, such as enzymatic catalysis, is also widely used in organic synthesis [14,15] due to the option to use mild reaction conditions, as well as its high catalytic activity and high selectivity. In an aqueous medium [15], fluorinase (5'-fluoro-5'-deoxyadenosine synthase (FDAS) EC 2.5.1.63)—derived from the bacterium *Streptomyces cattleya*—has been reported to catalyze the synthesis of 5'-fluoro-5'-deoxyadenosine (5'-FDA) using inorganic fluoride, such as KF or NaF, as a fluorine source. This fluorinase is also readily available through overexpression in *Escherichia coli*, which is used as a well-established cell factory and the most popular expression platform [16]. This enzyme can also catalyze the synthesis of 5'-[18F] FDA with [18F] fluoride as a nucleophile [17], which is among the most important radioisotopes in the radiopharmaceutical industry because of its long half-life ($t_{1/2}$ = 110 min) and is used in medical imaging and diagnostics such as positron emission tomography (PET), computerized tomography (CT), and magnetic resonance imaging (MRI) [18,19]. A major stumbling block to scale the application of the free enzyme includes disadvantages such as high associated costs and easy inactivation.

In contrast, the immobilized enzyme shows several distinct advantages [20,21] as a type of recyclable enzyme catalyst. It has highly efficient catalytic activity and has higher stability than the free enzyme under severe catalytic conditions. Among the strategies used to obtain the immobilized enzyme preparations, enzyme–inorganic hybrid nanoflowers (hNFs) may be the more conveniently prepared [22] because of the facile procedure and their enhanced activity and stability [22,23], which was inspired from biomineralization and contributes to the green development of chemical enterprises [24]. Currently, the inorganic metal ions used to prepare hNFs have also been expanded from $Cu^{2+}$ to $Zn^{2+}$, $Co^{2+}$, $Ca^{2+}$, $Mn^{2+}$, and so forth. [25–28]. Further, biomimetic mineralization has been developed as a convenient method to prepare biomimetic materials [29,30] such as artificial bone. In this process, fluoride can participate in many phases of calcium phosphate crystallization in vivo and exerts an important influence on this process [31]. This can be achieved by squeezing out the water due to the better affinity of fluoride and regulating the ionic composition around the mineral, thus strengthening the matrix protein–mineral interaction [32], which further affects the nature and physicochemical properties of the crystalized mineral. When fluoride ions were used as additives and doped on the biomimetic amorphous calcium phosphate (ACP), the obtained materials presented heightened remineralizing properties. It has also been verified that low doses of fluoride (0–2 ppm) in calcium phosphate solutions can inhibit the formation of octacalcium phosphate (OCP) and induce the growth of hydroxyapatite (HAp) formation [33].

In this work, to investigate the effect of fluorine (F) on the co-crystallization and biomineralization of fluorinase (5'-fluoro-5'-deoxy adenosine synthetase (FDAS)) and reusable calcium phosphate [34] to achieve the sustainable circular bioeconomy [35], the formation of nanocrystals of fluoride-substituted hydroxyapatite was mediated by fluoride and recombinant fluorinase to obtain fluorinase@fluoridated hydroxyapatite nanoflowers. The nanoflower was used in the biosynthesis of 5'-fluorodeoxy adenosine (Scheme 1), and the effects of reaction conditions on the yield were examined. Under the optimal crystallization conditions, flower-shaped hybrid catalyst containing F ions, different from fluorinase@hNFs are formed, not only presents the enhanced thermal stability, but also gives the conversion rate of the substrate of 98%.

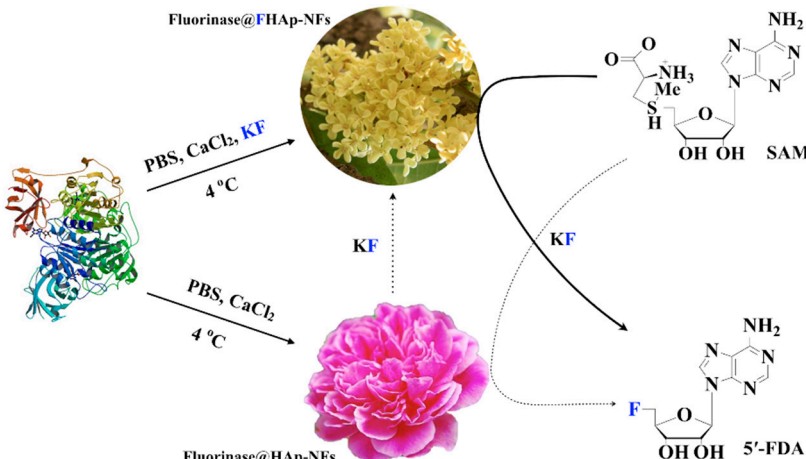

**Scheme 1.** Scheme for the bioinspired formation of fluorinase@fluoridated hydroxyapatite nanoflowers and enzymatic synthesis of the 5′-fluoro-5′-deoxyadenosine.

## 2. Materials and Methods

### 2.1. Bacterial Strains, Plasmids, Medium, and Chemicals

In gene cloning and recombinant expression of protein, *Escherichia coli* strains DH5$\alpha$, BL21 (DE3), Rosetta (DE3), and Origami (DE3) (TransGen Biotech, Beijing, China) were used. Plasmid pET28a (Novagen™, Merck, Darmstadt, Germany) was used as the vector for expressing *fla* gene (GenBank: DL242155) from *Streptomyces cattleya*. All standard recombinant DNA products were synthesized or purchased from Shanghai Generay Biotech Co, Ltd, Shanghai, China. Isopropyl-b-D-thioga lacto pyranoside was purchased from Sangon Biotech. S-adenosyl-L-methionine (SAM) was obtained from Sigma-Aldrich (St. Louis, MO, USA). Potassium fluoride was purchased from J&K Scientific Ltd. (Beijing, China). All other chemicals were analytical grade and were provided by Sinopharm Chemical Reagent Lt. D (Shanghai, China).

### 2.2. Gene Expression and Enzymes Protein Preparation

The putative *fla* gene encoding (GenBank: DL242155) was synthesized and subcloned into the expression pET28a (+) plasmid vector between restriction sites NdeI and XhoI. The recombinant plasmid pET28a-flA was used for expressing N-terminal His-tagged fusion. The pET28a-flA were transformed into *E. coli* BL21 (DE3) cells, and the *E. coli* BL21 (DE3) cells containing the pET28a-flA gene were incubated in Luria–Bertani medium (LB medium; 10 g tryptone, 5 g yeast extract, and 10 g NaCl dissolved in 1L of purified water) containing 50 μg·mL$^{-1}$ kanamycin at 37 °C with shaking. The *E. coli* BL21(DE3) cells containing pET28a-flA were incubated in 100 mL of LB medium until the OD$_{600nm}$ reached approximately 0.6–0.8. Sequentially, isopropylthiogalactoside (IPTG) of approximately 0.5 mM (final concentration) was added to LB medium promptly, and the induction was carried out at 18 °C for approximately 12 h.

Cells were harvested by centrifugation (8000× $g$ for approximately 5 min at 4 °C) and were washed by resuspending three times and were then resuspended in 20 mM sodium phosphate buffer (PBS) (pH 7.5). The sonication protocol included 3 s sonication/7 s rest for a total of 10 min on ice using a Sonicator 400 (Misonix, Farmingdale, NY, USA). After centrifugation, the supernatant was filtered by a 0.45 μm filter (Millipore, Burlington, MA, USA), while the cell lysate was purified by a column packed with Ni$^{2+}$-charged His-Bind Resin (Ni-NTA, Bio Basic™, Markham, ON, Canada). Resin-bound recombinant protein was eluted with 20 mM sodium phosphate buffer (pH 7.5) containing 250 mM imidazole, 0.5 M NaCl, and 10% glycerol. Then, by dialysis at 4 °C for 12 h, it was desalted three times using deionized water. The expressed and purified recombinant FDAS was analyzed by sodium dodecyl sulfate–polyacrylamide gel electrophoresis (SDS-PAGE, 12% *w/v* acrylamid). A Bradford

protein assay kit (Quick StartTM, Bio-Rad, Berkeley, CA, USA) was used to determine the concentration of protein.

### 2.3. Co-Expression of flA and Chaperone Genes in E. coli

When optimizing the inducer concentration, the recombinant *E. coli* BL21f harboring pET-28a-flA were inoculated at 37 °C and grown in 10 mL of LB medium containing 50 μg·mL$^{-1}$ of kanamycin until the OD$_{600}$ reached approximately 6. IPTG (final concentration, 0.1mM, 0.5 mM, 2.0mM) was added immediately to induce the expression of flA to produce FDAS at 18 °C for 12 h. The cells were also harvested by centrifugation (8000× *g* for approximately 5 min at 4 °C), washed three times, and then resuspended in 3 mL of 20 mM phosphate buffer (pH 7.5). After disruption by sonication (Sonicator 400, Misonix, Farmingdale, NY, USA), the supernatant fraction was separated from the cell lysate by centrifugation at 4 °C and 12,000× *g* for 20 min. The optimal induction concentration of IPTG was determined by SDS-PAGE electrophoresis analysis. The induction temperature (16 °C, 18 °C, 23 °C) and time (6 h, 12 h, 24 h) were also optimized for the expression of *flA*.

To avoid the formation of inclusion in this gene expression, transformation of *E. coli* with pGro7 or pKJE7 (Chaperone Plasmid, Takara, Japan) plasmid was followed by transformation with an expression plasmid for the target flA. To perform co-expression of these two genes, the transformant was inoculated into medium containing 20 μg·mL$^{-1}$ chloramphenicol for selecting plasmid and 0.5–4 mg·mL$^{-1}$ L-arabinose for inducing chaperone expression. BL21 (DE3), Rosetta (DE3), and Origami (DE3) were used to screen the suitable expression host for the flA gene. When harboring the flA gene, they were grown in 100 mL LB media supplemented with 100 μg·mL$^{-1}$ chloramphenicol or streptomycin, then incubated at 37 °C. After culturing, the amount of soluble target protein in the supernatant was determined by SDS-PAGE.

### 2.4. Preparation and Characterization of Enzyme@fluoridated Hydroxyapatite Nanoflowers (FHAp-NFs)

Enzyme@fluoridated hydroxyapatite nanoflowers (FHAp-NFs) were synthesized as described in a previous study [36] and in our previous work [37]. A total of 2 mL of CaCl$_2$ solution (0.2 mol·L$^{-1}$) was added to 100 mL of a phosphate-buffered saline (50 mmol·L$^{-1}$, pH 6.5, 7.5, 8.5) solution containing 5, 20, 70, 100 μg·L$^{-1}$ FDAS and a certain concentration (0.5 ppm, 1 ppm, 5 ppm, 20 ppm, 50 ppm, and 3000 ppm) of KF, respectively [33]. The mixture was then maintained at 4 °C for different periods of time (0, 4, 12, 24 h) for co-crystallization of the enzyme and fluorapatite. Then, the mixture was centrifuged at 4 °C and 10,000 rpm for 10 min, and the enzyme–inorganic fluoridated hydroxyapatite nanoflowers (FHAp-NFs) were obtained as a precipitate by removing the supernatant.

The centrifuged pellet was washed three times with ultrapure water and lyophilized for the following tests. A scanning electron microscope (SEM, S4800, Hitachi High Technologies Corporation, Tokyo, Japan) was used to observe the morphology of FHAp-NFs, and secondary electron images were acquired by a field emission scanning electron microscope at 3 kV and a current of 5 mA. Powder X-ray diffraction (XRD, Bruker D8 Advance X-ray Diffraction) was also used to examine and confirm the crystal structures and components of the FHAp-NFs. The FT-IR spectra of the FHAp-NFs were recorded using a FT-IR spectrophotometer (Thermo Nicolet iS5) from 2000 to 500 cm$^{-1}$ with samples dispersed in KBr pellets.

### 2.5. Activity Assay and General Procedure of Enzymatic Synthesis of 5′-Fluorodeoxy Adenosine (5′-FDA)

Enzymatic activity for fluorinase preparations was assayed by a 5′-FDA synthesis reaction. This product was monitored, analyzed, and confirmed by high-performance liquid chromatography (HPLC) using the synthetic pure 5′-FDA as a standard at 25 °C. Protein fractions (1 mg·mL$^{-1}$) were incubated at 37 °C with SAM (2 mM) and KF (50 mM) in Tris-HCl buffer (50 mM, pH 7.5) in a final volume of 2 mL [37]. After boiling at 95 °C (3 min) and removing the precipitated proteins by centrifugation, the samples were then used for HPLC analysis. The clarified supernatant was filtered through a 0.22 μm filter membrane and used for high-performance liquid chromatography (HPLC, Agilent 1260) on a C18

column (250 mm × 4.6 mm, particle size 5 μm) equilibrated with $KH_2PO4$ (50 mM) and acetonitrile (99:1 *v/v*). The pump was run at a flow rate of 1 mL·min$^{-1}$, and the injection volume was 10 μL. The oven was maintained at 30 °C, and data were collected at 254 nm; retention times were verified using standard solutions of substrates and products. All experiments were performed in triplicate. A 5′-Fluoro-5′-deoxyadenosine standard was synthesized as the report [13] and characterized using $^{19}$F nuclear magnetic resonance (NMR) spectra and gas chromatography–mass spectrometry (GC–MS, Agilent 5975).

In order to determine the kinetic constant of KF, the effects of different KF concentrations on the product 5′-FDA were tested under certain SAM concentrations. The 5′-FDA production concentration was calculated according to the standard curve of the product, and a kinetic curve of the 5′-FDA production rate was obtained at this KF concentration. The kinetic constant of KF was obtained based on the Michaelis–Menten equation.

### 2.6. Optimum Temperature, pH, and Reaction Time of Enzyme Preparations

For optimization of the reaction temperature and pH in medium, enzyme preparations containing 1 mg of fluorinase were incubated with 2 mM SAM and 50 mM KF in 50 mM potassium phosphate buffer, and the total reaction volume was 1 mL. The activities of fluorinase preparations were examined in 50 mM potassium phosphate buffer (pH 7.5) at different temperatures, ranging from 20 to 60 °C, to identify the optimum reaction temperature. To detect the effect of pH on the reaction, the effect of pH on the activities of fluorinase preparations was determined at 37 °C in different pH buffers. The 0.05 M buffers including citric acid–sodium citrate buffer (pH 4.0–6.0), potassium phosphate (pH 6.0–8.5), and glycine–sodium hydroxide buffer (pH 8.0–10.0) were used, and all experiments were performed in triplicate.

In order to choose a suitable reaction time with a total reaction volume of 10 mL, 10 mg of FDAS and FDAS@FHAp-NFs were incubated for a certain period with 2 mM SAM, 50 mM KF, and 50 mM potassium phosphate buffer (pH 7.5) at 37 °C [15]. At regular intervals, a 500 μL aliquot was withdrawn from the reaction mixture and mixed with 500 uL of acetonitrile. The product content was determined using the HPLC standard assay method, and all experiments were conducted in triplicate.

### 2.7. Thermal Stability of Fluorinase Preparations

The thermal stability of recombinant FDAS, FDAS@HAp-NFs, and FDAS@FHAp-NFs were investigated by incubation of 10 mg purified enzyme in 10 mL and 20 mM potassium phosphate buffer (pH 7.5) at 30 °C, 40 °C and 50 °C for approximately 8 h. At intervals of 1 h, 500 μL aliquots were taken from the incubation mixture, and 500 uL of acetonitrile was added. The residual activities of enzyme preparations were measured using the standard assay method. The initial activities of enzyme preparations were all designated as the control. All experiments were conducted in triplicate.

## 3. Results and Discussion

### 3.1. Expression and Purification of the Recombinant Fluorinase

A gene fragment of 909 bp encoding f1A was cloned and inserted into the vector pET28a, and *E. coli* BL21 (DE3) with the recombinant plasmid pET28a-f1A was cultured in LB medium, after which the expression of fluorase was induced. The expression of recombinant f1A was analyzed by SDS-PAGE, and the results in Figure 1A show that the molecular weight of the recombinant f1A protein is approximately 34.4 kDa, which is consistent with that calculated from the amino acid sequence [38]. For the low levels of the *flA* gene expression, it is necessary to optimize the expression conditions in *E. coli* BL21. However, the concentration of IPTG has little effect on the soluble target protein expression, as the results showed (Figure S1, ESI). With the progress of the IPTG concentration, the total protein expression for the whole cell increased, and the insoluble matter also increased. Additionally, when the temperature was 16 °C (Figures S2 and S3, ESI), there was less induced insoluble matter than at 18 °C.

Similarly, the induced soluble target protein decreased with declining temperature. The 12 h period was suitable for the induction (Figure S4, ESI).

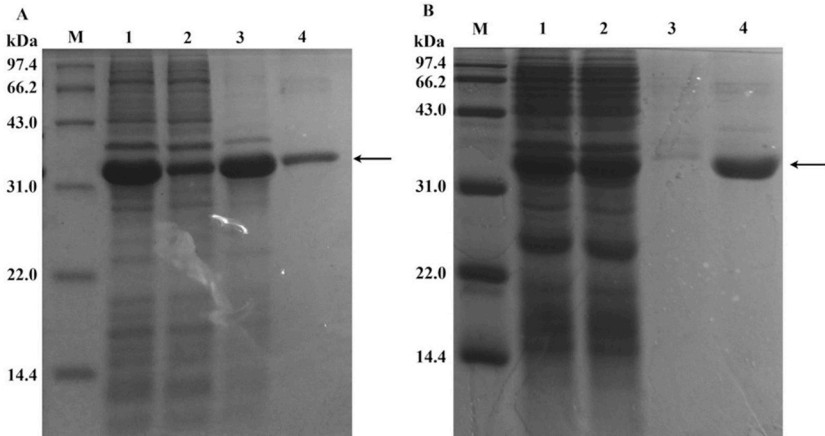

**Figure 1.** SDS-PAGE analysis of the recombinant fluorinase expressed in different hosts ((**A**), BL21; (**B**), Rosetta). Lane M, protein molecular mass marker; Lane 1, whole-cell proteins; Lane 2, soluble region; Lane 3, insoluble region; Lane 4: purified recombinant flA.

In cells, molecular chaperones are a class of proteins that assist intracellular molecular assembly and protein folding [39]. One of their main functions is to recognize and stabilize the partially folded conformation of the polypeptide chain, thereby participating in the folding and assembly of the new peptide chain. In addition to the basic role of assisting protein folding, they can also participate in protein transport, assembly, aggregation, and degradation, which aids the nascent polypeptides to reach their final structure [36,40]. However, as shown in Figure S5, molecular chaperones have not assisted and improved the correct expression of target protein when the commercial chaperone plasmid set pGro7 from Takara was used in *E. coli* BL21 (DE3). Moreover, the pKJE7 chaperone decreased and inhibited the normal expression of fluorinase.

The reason for the above phenomenon may be that there are significant differences in the frequency of using different codons between different organisms, although the expression hosts and operons are the same [41]. As shown in Figure S6 (ESI) and Figure 1B, when the expression host is Rosetta (DE3), the content of insoluble expressed fluorinase significantly decreases. Generally, the concentration of protein was 3 mg·mL$^{-1}$, which was higher than that in *S. cattleya* NRRL 8057 [37]. The reason may be that the recombinant plasmid pET28a-flA belongs to a heterologous expression, and the expressed host cell is completely different from the previous one. Recombinant proteins that are not expressed or truncated by the host due to codon preference force the host to express specific proteins, and it does not have a large number of tRNAs. As a result, the recombinant gene cannot be expressed, or the expression level is low, and the activity of the expressed enzyme is reduced. Recombinant plasmids with lower expression levels generally include rare codons. For the heterologous expression of our recombinant plasmid pET28a-flA to achieve overexpression, the biggest problem to be solved is the solution of rare codons. The flA gene is derived from fungi and contains rare codons. The flA gene is expressed using *E. coli* as the host cell, and compared with typical *E. coli* proteins, the amino acid composition of the corresponding protein is biased. Thus, it may face translation problems, including translation pauses, premature translation termination, and translation frameshifting to reduce expression of the protein's quantity or quality. The tRNA genes of the engineering strain *E. coli* Rosetta include rare codons AGG, AGA, AUA, CUA, CCC, and GGA. Therefore, codon usage optimization is critical in *E. coli* [42,43].

### 3.2. Preparation and Characterization of Enzyme@FHAp-NFs

Prior to preparing the fluorinase@fluoridated hydroxyapatite nanoflowers (FDAS@FHAp-NFs), we examined the effect of metal ions on fluorinase. Fluorinase was used without any metal ions and

EDTA as the control group, and its catalytic synthetic activity was set as 100%. As shown in Figure 2, 1.0 mM $Ca^{2+}$, $Mn^{2+}$, and $Co^{2+}$ exerts no influence on the activity of FDAS.

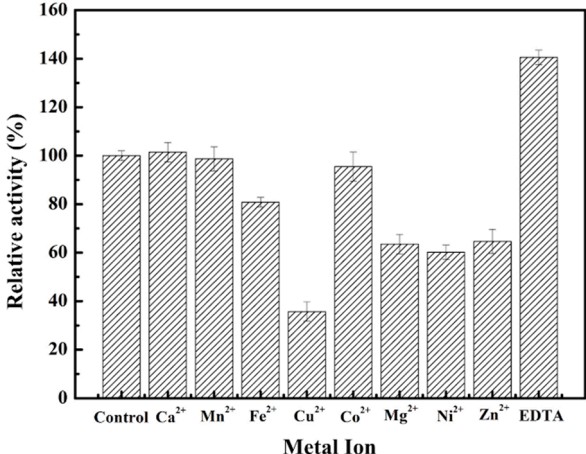

**Figure 2.** Effects of metal ions and EDTA on the FDAS (5′-fluoro-5′-deoxy adenosine synthetase) synthetic activity.

Hydroxyapatite (HAP), fluorinase@hydroxyapatite (FDAS@hNFs), fluoride-substituted hydroxyapatite (FHAp), and fluorinase@fluoridated hydroxyapatite nanoflowers (FDAS@FHAp-NFs) were characterized by SEM, and the results are presented in Figure 3. The formation of FDAS-hydroxyapatite is based on the mechanism of biomineralization. The specific process that occurs is the regulation and influence of the biological macromolecular fluorinase protein, calcium ions, phosphates, and amino and carboxyl groups on the fluorinase protein, which work together to trigger the formation of nanocrystalline nuclei. With the gradual increase of crystal nuclei, lots of protein molecules and primary crystals are agglomerated, and finally, a protein–hydroxyapatite complex is formed, which has a shape similar to a petal [44]. For the formation of fluorinase@fluoridated hydroxyapatite nanoflowers, the fluoride ions added to the solution can remineralize with apatite crystals and biomacromolecules in several different ways in the presence of their different concentrations and solution compositions. At low fluoride levels, F-OH exchange between apatite solution phases can easily occur. By comparing the structures of the fluorinated hydroxyapatite and the hydroxyapatite, it can be seen that the bond length of the fluorinated hydroxyapatite is shorter than that of the hydroxyapatite, which means that the FDAS-FHAp-NFs formed is more stable. As time passes, more consistent and regular FDAS-FHAp-NFs are produced, leading to the formation of a more complete structure, but with a shape different from the FDAS-HAp-NFs structure [32].

In Figure S7A–C (ESI), the effects of pH and crystallization time in the different buffers on the crystal sHApe are shown. It is apparent that pH has a great influence on the crystal form, and the crystal form is more regular at pH 7.5. Crystal growth is slow and may be affected by ambient temperature, since the enzyme activity of the fluorinase is also sensitive and greatly affected by the ambient temperature. To prevent the structure and activity of this enzyme from being damaged, the temperature of the long co-crystallization was selected to be 4 °C, and the time required for the entire bionic immobilization of enzyme protein is 24 h (Figure S7D–E, ESI).

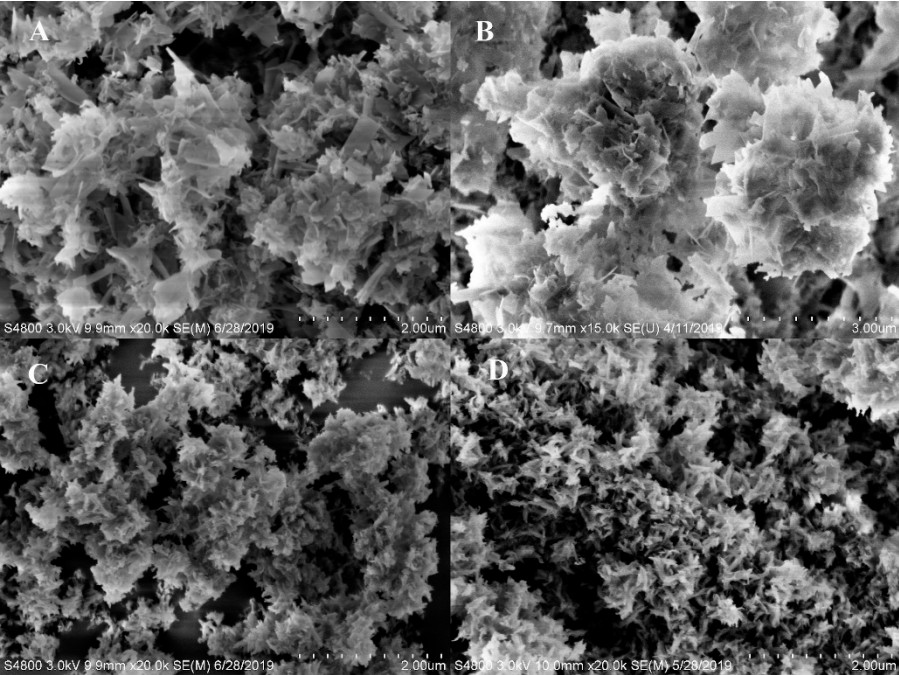

**Figure 3.** SEM photographs of hydroxyapatite (HAp, (**A**)), fluorinase@hydroxyapatite nanoflowers (FDAS@HAp NFs, (**B**)), fluoridated hydroxyapatite (FHAp, (**C**)), and fluorinase@fluoridated hydroxyapatite nanoflowers (FDAS@FHAp-NFs, (**D**)).

Figure S8 (ESI) presents the effect of fluorinase concentration on crystal growth. It can be seen from the SEM photograph that as the concentration of the enzyme gradually increases, the overall structure of the crystal gradually increases and becomes regular. When the enzyme concentration is not less than 70 µg·mL$^{-1}$, fluorinase@fluoridated hydroxyapatite nanoflowers form is preferred more than at higher enzyme concentrations. When the concentration of enzyme is too high, the enzyme loading rate on the support will decrease. Figure S9 (ESI) shows the effect of different F ion concentrations on the crystal formation. When the F ion concentration is not less than 20 ppm, the crystal form is relatively regular, but when the F ion concentration is as high as 3000 ppm, spherical calcium fluoride crystals are more easily formed. This may result from the biomineralization process that takes place under the control of organic matter, and the low concentration of organic matter would result in less regulation.

Figure 4 displays the XRD analysis of the enzyme@FHAp-NFs, F-HAP, enzyme@hNFs, and HAP. The XRD analysis of fluoridated hydroxyapatite nanoflowers confirmed the opinion, while the relative strength of each diffraction peak of hybrid nanocomposites matched the standard pattern of fluoride-substituted hydroxyapatite ($Ca_5(PO_4)_3F$, FHAP), octacalcium phosphate ($Ca_8H_2(PO_4)_6 \bullet 5H_2O$, OCP), hydroxyapatite ($Ca_{10}(OH)_2(PO_4)_6$, HAp), $CaHPO_4 \bullet 2H_2O$, and its other derivatives. The wide-angle XRD pattern of the enzyme, FHAp-hNFs, shows distinct diffraction peaks at 11.72°, 26.03°, 32.12°, 32.29°, 40.65°, 46.85°, 49.65°, 53.25°, which were the most prominent characteristic peaks of $Ca_5(PO_4)_3F$ and $Ca_{10}(OH)_2(PO_4)_6$. However, the diffusion peak at 2θ = 31°–34° may demonstrate the presence of amorphous calcium phosphate [31]. In addition, this broadening peak could also result from the appearance of smaller apatite crystals, which also include octacalcium phosphate ($Ca_8H_2(PO_4)_6 \bullet 5H_2O$, OCP). Evidently, it can be seen from 2θ = 30°–34° that the crystal changes to $Ca_5(PO_4)_3F$ after the addition of F ions when preparing the enzyme@FHAp-NFs and enzyme@hNFs. The presence of $Ca_5(PO_4)_3F$ in the pattern of enzyme FHAp-NFs can be easily verified by peaks at 31.87°, 32.2°, 39.15°, 39.98°, and 50.63° [31,44].

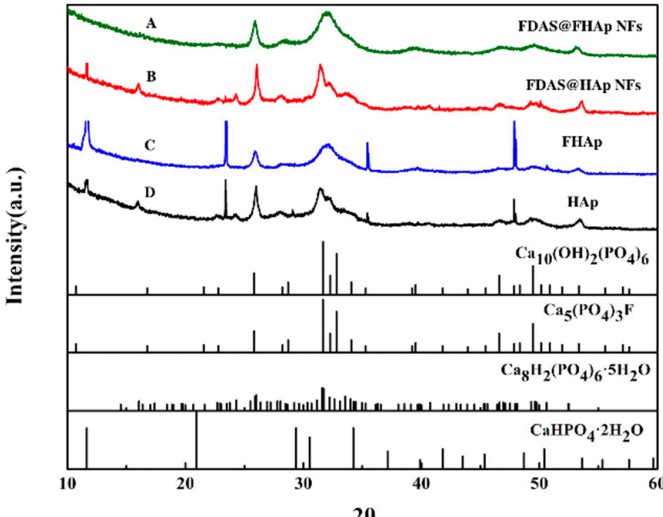

**Figure 4.** XRD of fluorinase@fluoridated hydroxyapatite nanoflowers (FDAS@FHAP NFs, (**A**)), fluorinase@hydroxyapatite nanoflowers (FDAS@HAp NFs, (**B**)), fluoridated hydroxyapatite (FHAp, (**C**)), and hydroxyapatite (HAp, (**D**)).

FT-IR was also used to characterize the obtained fluorinase@fluoridated hydroxyapatite nanoflowers, and the result in the upper panel in Figure 5 indicates that the remineralized complex, which was prepared using fluorinase, presents protein and phosphate minerals. The main absorption peak of the protein is the amide band, including band I, band II, and band III. Amide band I (1690–1630 cm$^{-1}$) is mainly due to the strong absorption of $\gamma_{C=O}$. Comparing this with that of the free enzyme, a red shift occurs to the amide of the immobilized enzyme. It is speculated that the enzyme may adhere to the inorganic support hNFs, which have some influence on the spatial structure of the enzyme. The amide band II (1420–1400 cm$^{-1}$) is the absorption band of the strong carbon–nitrogen bond stretching vibration, and the absorption band of the amide bond of the enzyme on the support shifted a little, which indicates that no change of the main structure occurs in the enzyme protein after the immobilization by biomineralization and the formation of h-NFs. In addition, the $(PO_4)^{3-}$ belt and the $\gamma_4$ belt are absorption bands for the $(PO_4)^{3-}$ stretching vibration. At the same time, the $\gamma_4$ band (600 cm$^{-1}$ and 558 cm$^{-1}$) demonstrates the interaction of calcium phosphate with the fluorinase protein after biomimetic mineralization [31]. Comparisons of the FT-IR spectra of samples A, B, C, and D show that the shift of the bands will also be offset after the introduction of F. This shift may be the overlap of fluoridated hydroxyapatite and octacalcium phosphate (OCP), when the hydroxyapatite layer is grown on the OCP precursor [45].

### 3.3. Optimum Temperature and pH of Free FDAS and FDAS@ FHAp-NFs Activity

A total of 2 mL of reaction mixture containing 2 mg of FDAS, 2 mM SAM, 50 mM KF, and 20 mM potassium phosphate buffer (pH 7.5) was incubated at 37 °C for 0.5 h. Figure 6A shows that the optimum temperature for enzyme preparations is 37 °C, which is similar to that of the free fluorinase from the marine-derived bacteria *Streptomyces xinghaiensis* NRRL B24674 [46]. Importantly, fluorinase@fluoridated hydroxyapatite nanoflowers present a wider suitable temperature range for the catalytic reaction and approximately 4.8-fold higher catalytic activity than free fluorinase. The reason for this may be that F ions are added during the biomineralization process and F ions are also the raw materials required for the synthetic reaction. Moreover, the substrate, S-adenosyl-L-methionine (SAM), is a polyamino and polyhydroxy compound, and the support, apatite, is also full of hydroxy, which can form the hydrogen bond network by the F$^-$ [9,33] (Figure 6A). This network would enrich the substrate SAM around the support and enzyme, which further enhances the catalytic activity of fluorinase.

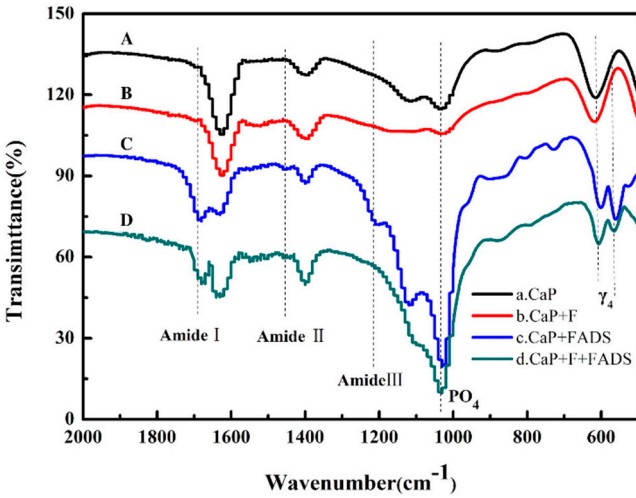

**Figure 5.** FT-IR spectra of hydroxyapatite (HAp (**A**)), fluoridated hydroxyapatite (FHAp (**B**)), fluorinase@hydroxyapatite nanoflowers (FDAS@HAp NFs (**C**)), and fluorinase@fluoridated hydroxyapatite nanoflowers (FDAS@FHAP NFs (**D**)).

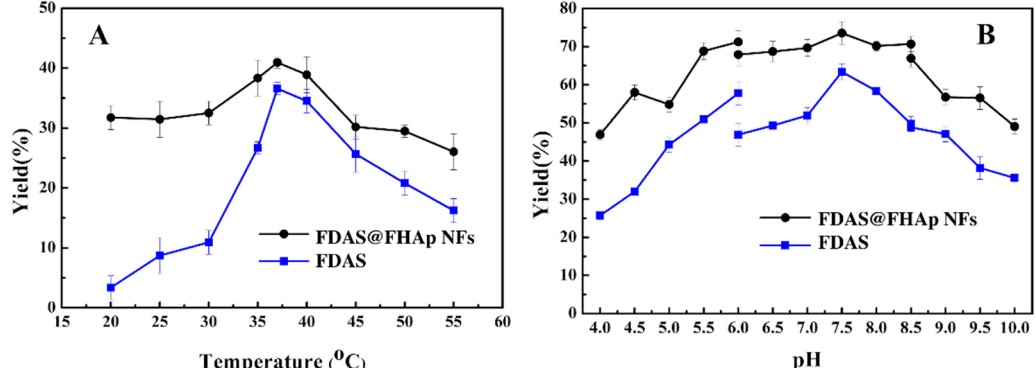

**Figure 6.** Optimum temperature (**A**) and pH (**B**) profiles for the synthetic activity of FDAS@FHAp NFs and FDAS. Three different buffers, citric sodium citrate buffer (pH 4–6), phosphate buffer (pH 6–8.5), and glycine/sodium hydroxide buffer (pH 8.5–10) were used.

Figure 6B shows the optimum pH of the enzyme preparations. The optimal pH of both free FDAS and FDAS@FHAp-NFs activity was approximately pH 7.5, and enzyme protein fractions (1 mg·mL$^{-1}$) were incubated at 37 °C with SAM (2 mM) and KF (50 mM) in different buffers and in a final volume of 2 mL for 12 h. When compared with a free enzyme over a broad range of pH (from pH 5.5 to 8.5), FDAS@FHAp-NFs preparation demonstrated greater stability. Additionally, FHAp-NFs present more activity than free enzymes in most pH conditions. However, the free fluorinase from *Streptomyces xinghaiensis* NRRL B24674 would undergo a remarkable decrease of activity in the catalytic synthesis under basic conditions (pH 8.0–9.5) [46].

### 3.4. Thermal Stability of Free FDAS and FDAS@FHAP

To evaluate the effect of temperature on enzyme catalytic activity, the enzyme preparations were incubated at temperatures ranging from 30 to 50 °C for different time periods. As indicated in Figure 7, the FDAS-FHAp-NFs proved to be more thermostable than the free enzyme, and approximately 80% of the initial activity remained after heating for 8 h at 30 °C. After heating at 50 °C for 8 h, more than 60% of the initial activity was retained. However, both the FDAS-HAp-NFs and free enzyme lost most of their initial activities. Obviously, an enzyme with support presents better stability than the free enzyme. When the enzyme was confined in the fluoridated hydroxyapatite, the thermal stability of fluorinase

was further improved. That is to say, compared with hydroxyapatite (HAp), fluoride-substituted hydroxyapatite (FHAp) was more suitable for fluorinase to retain its catalytic activity.

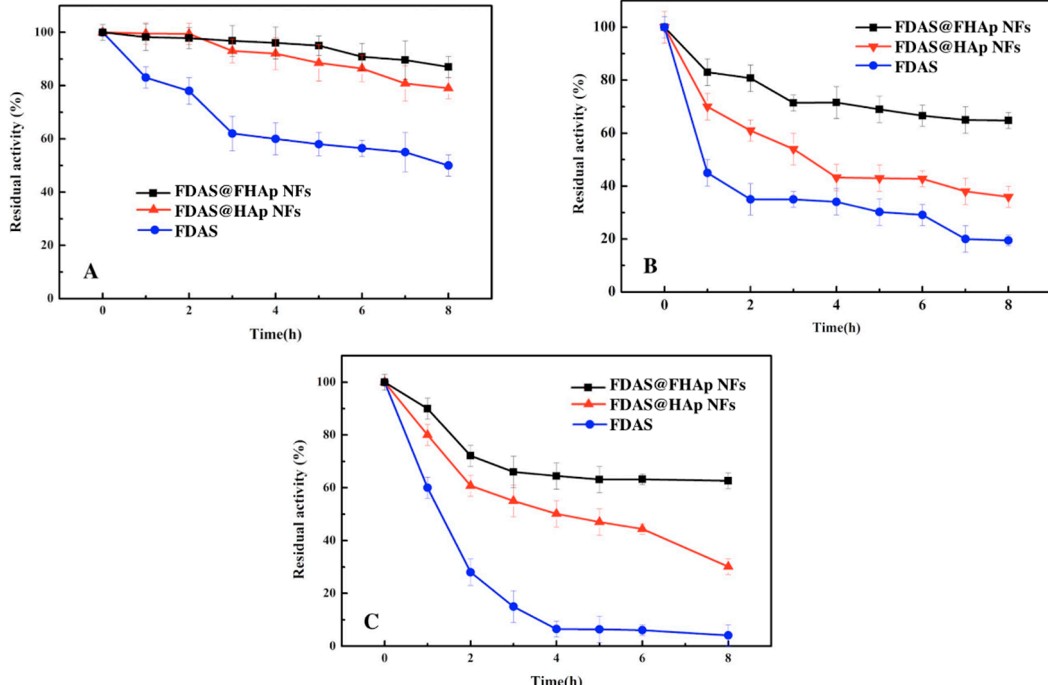

**Figure 7.** Thermal stability of FDAS, fluorinase@hydroxyapatite nanoflowers (FDAS@HAp NFs), and fluorinase@fluoridated hydroxyapatite nanoflowers (FDAS@FHAP NFs) at 30 °C (**A**), 40 °C (**B**), and 50 °C (**C**).

The enhanced thermal stability of the FDAS-FHAp-NFs may be due to the substitution of $F^-$ for $OH^-$ on the support, which results in a reduction of the unit cell volume and a denser lattice. Subsequently, the electrostatic attraction between fluoride and adjacent ions would strengthen, and the thermal stability of the FDAS-FHAp-NFs would be greatly improved. Fluoride has been shown to be able to replace the columnar hydroxyl groups, which are distributed in the apatite structure [32]. This reduces the volume of the formed crystals and, therefore, increases the rigidity of the confined enzyme and consequently, enhances the thermal stability of the enzyme preparation.

### 3.5. Synthesis of 5′-Fluorodeoxy Adenosine using FDAS Enzyme Preparations

Based on the above experimental research, FDAS preparations were used to catalyze the synthesis of 5′-fluorodeoxy adenosine with S-adenosyl-L-methionine as substrate at 37 °C (Figure 8). It can be seen from Figure 8 that the peak position of the substrate SAM is 2.3 min, and the peak position of the product is 3.2 min. Moreover, at the optimum pH of 7.5, the activity of FHAp-NFs was approximately 2-fold higher than that of the free enzyme.

To examine the effect of reaction time on the yield of target product 5′-fluorodeoxy adenosine (5′-FDA), at regular intervals, a 500 μL sample was taken from the incubation mixture and used for HPLC analysis. The reaction mixture was incubated at 37 °C with SAM (0.8 mM), KF (10 mM), and fluorinase (1 mg·mL$^{-1}$) in a total volume of 10 mL. The results derived from Figure S10 show that the observed Km for $F^-$ is 1.12 ± 0.21 mM and kcat/Km is 0.13 mM$^{-1}$·min$^{-1}$. However, the Km for $F^-$ is 10 ± 2 mM and kcat/Km is only 0.059 mM$^{-1}$·min$^{-1}$ for the free enzyme [47]. Therefore, the catalytic efficiency of the FDAS@FHAp-NFs is two-fold higher than that of the free enzyme.

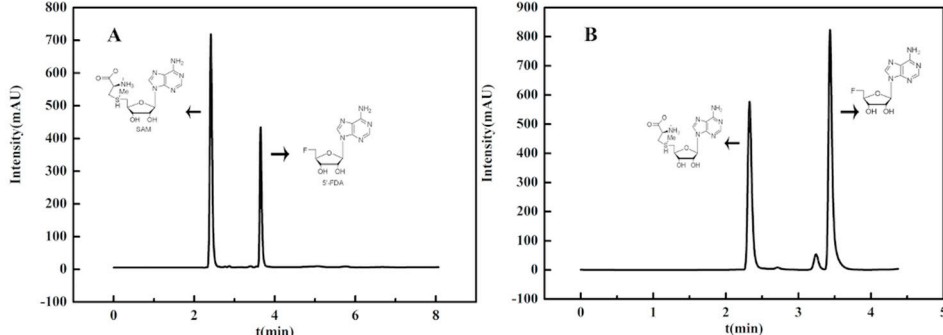

**Figure 8.** Comparison of HPLC analysis of the synthesized 5′-fluorodeoxy adenosine using FDAS enzyme preparations ((**A**) free FDAS catalysis and (**B**) enzyme@FHAp-NFs of FDAS catalysis).

As shown in Figure 9, more than 98% of S-adenosyl-L-methionine (SAM) substrate was transformed to product 5′-fluorodeoxy adenosine using FDAS@FHAp-NFs after 25 h of reaction. In addition, the FDAS@FHAp-NFs gave a better catalytic activity than that of the same amount of free FDAS in the synthesis of 5′-fluorodeoxy adenosine (5′-FDA) when S-adenosyl-L-methionine (SAM) and a fluoride ion were used as substrates. By comparison, it can be concluded that in the first two hours, the conversion rate of the immobilized enzyme was twice that of the free enzyme. With the extension of the reaction time, the product conversion rate gradually increases, but the free enzyme is still lower than the immobilized enzyme. This may result from the more remarkable loss in enzyme activity for the free enzyme than for the FDAS@FHAp-NFs as the reaction time increases.

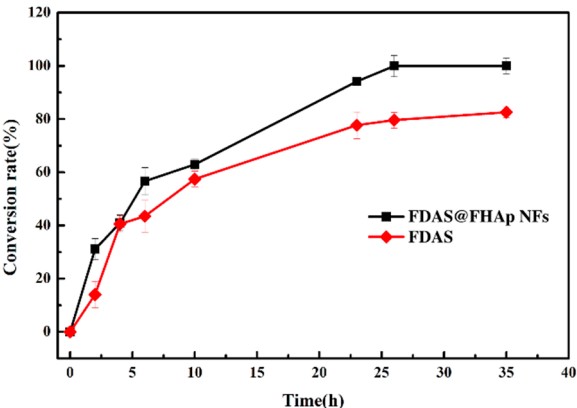

**Figure 9.** Effects of reaction time on the synthesis of 5′-FDA using fluorinase@hydroxyapatite nanoflowers (FDAS@HAp NFs).and fluorinase (FDAS) as biocatalysts.

## 4. Conclusions

In conclusion, a facile method was reported to prepare the enzyme@FHAp-NFs, which can be further used for capable enzymatic synthesis of 5′-FDA. FDAS promotes the formation of directional beams of flower-like fluorinated hydroxyapatite in a biomineralized manner. Confined in the FHAp to form nanoflowers, FDAS presented superior catalytic activity and thermal stability, which may be due to the hydrogen bond network containing $F^-$ ions. The process of enzyme@FHAp-NFs preparation was simple and efficient, and it may be further applied to the immobilization of other enzymes for biocatalysis using hydrophilic substrate. In addition, immobilized fluorinase will play an increasingly important role in fluorine chemistry due to its excellent catalytic specificity and stability.

**Supplementary Materials:** The following are available online at http://www.mdpi.com/2071-1050/12/1/431/s1. Figure S1: Expression assays of fluorinase by different concentration of IPTG, Figure S2: Expression assays of fluorinase at 16 °C induction, Figure S3: Expression assays of fluorinase at 23 °C induction, Figure S4: Expression assays of fluorinase at different induction time, Figure S5: Expression assays of fluorinase with pGro7 or pKJE7,

Figure S6: Expression assays of fluorinase in different expression host, Figure S7: Effect of pH in PBS on crystal form, Figure S8: Concentration of FDAS in PBS, Figure S9: Concentration of different F⁻, Figure S10. Effect of substrate KF concentration on the reaction rate.

**Author Contributions:** Conceptualization, A.W. and T.X.; methodology, A.W. and N.L.; software, N.L. and H.L.; validation, N.L.; formal analysis, N.L. and T.M.; investigation, N.L. and B.H.; resources, T.X.; data curation, Y.Y.; writing—original draft preparation, N.L.; writing—review and editing, A.W.; visualization, N.L.; supervision, A.W.; project administration, A.W.; funding acquisition, T.X. All authors have read and agreed to the published version of the manuscript.

**Funding:** This study was supported by the National Natural Science Foundation of China (21576062, 81730108), Natural Science Foundation of Zhejiang Province (LY18B060009, LY15B060011), National Innovation and Entrepreneurship Training Program for Undergraduate (201810346008), Technology Research and Development Program of Hangzhou(20191203B09), National innovation and entrepreneurship project for College Students(201810346008, 201910346044), Scientific Research Innovation Fund for Graduate Students of Hangzhou Normal University, "Star and light" Project for Talent Students in Hangzhou Normal University.

**Acknowledgments:** We would like to convey a special thanks to Xiaojun Cao for SEM characterization of the nanoflowers samples.

**Conflicts of Interest:** The authors declare no conflict of interest.

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
