# Peer review of "Facile Bioinspired Preparation of Fluorinase@Fluoridated Hydroxyapatite Nanoflowers for the Biosynthesis of 5′-Fluorodeoxy Adenosine"

_sustainability, doi:10.3390/su12010431_

Round 1
Reviewer 1 Report
This paper deals with the development of a biocatalyst for the synthesis of organofluorine compounds. The authors conclude that the fluoride efficiently enhances the catalytic activity and thermal stability of fluorinase in the hybrid nanoflowers for the synthesis of 5’-fluorodeoxy adenosine.
Good paper; useful method and material. The proposed preparation method is quite interesting and well discussed. The properties of the samples are studied with a series of techniques in a very thorough and detailed way and the results appear very interesting.
The introduction need to be upgraded with more recent references on the topic.
The manuscript contains some lingual deficiencies (grammar, syntax) that have to be eliminated. English language should be improved.
Author Response
Reviewer #1:
Question 1:
The introduction need to be upgraded with more recent references on the topic.
Answer:
Thank you for your kind comments and suggestions.
We have revised introduction and the corresponding changes were marked in red. More recent references on the topic were also used to introduce the progress in the fluorinase and enzyme-inorganics hybrid nanoflowers (Int. J. Biol. Macromol. 2018, 118, 1369-1376.;ACS nano 2019, 13, 3151-3161).
Question 2:
The manuscript contains some lingual deficiencies (grammar, syntax) that have to be eliminated. English language should be improved.
Answer:
Thank you for your suggestions.
For grammar issues, we have revised the English in the manuscript and the corresponding proportions were all marked in red.

Reviewer 2 Report
The current manuscript deals with the preparation of hydroxyapatite nano flowers with fluorinase activities and their assessment for the synthesis of 5'-fluorodeoxy adenosine.The introduction is too long and will deserve to be more focus on the central point of the study. The enzymatic activities of the nanoflowers should also be better characterized. What about the Km and the kcat compared to the free fluorinase? What is the specific activity? Figure 9 is also poorly informative. What are the yield of conversion? What amount of compounds can we get from the procedure? Is there any improvement compared to the free fluorinase ?
Author Response
Reviewer #2:
Question 1:
The introduction is too long and will deserve to be more focus on the central point of the study.
Answer:
Thank you for your comments and suggestions very much.
According to your suggestion, the introduction has been revised and the corresponding proportions were marked with a red. The introduction is briefly divided into three parts, including the background of fluoride research, the development of fluoride enzymes and the background of immobilized enzymes.
Question 2:
The enzymatic activities of the nanoflowers should also be better characterized. What about the Km and the kcat compared to the free fluorinase? What is the specific activity?.
Answer:
Thank you for your suggestions.
In the references, determination of enzyme activity is performed by high performance liquid chromatography, which is the same method as the detection of products. The enzyme activity was characterized and calculated from the relative yield of the product.
For the immobilized enzyme, the results derived from the Fig.S10 show that the observed Km for F- is 1.12±0.21mM and kcat/Km is 0.13 mM-1·min-1. However, the Km for F- is 10±2 mM and kcat/Km is 0.059 mM-1·min-1 for free enzyme (J. Am. Chem. Soc. 2007, 129, 14597), respectively. Therefore, the catalytic efficiency of the FDAS@FHAp-NFs is higher than that of free enzyme..
Question 3:
Figure 9 is also poorly informative.
Answer:
Thank you for your suggestions.
According to your requirements, I have added the analysis in Figure 9 in the revised manuscript.
Fig. 9 mainly presents the comparison of the reaction time required for the immobilized enzyme and free enzyme under proper catalytic conditions. The results show that in the first two hours, the conversion rate of the immobilized enzyme was twice that of the free enzyme. With the extension of the reaction time, the product conversion rate is gradually approaching, but the free enzyme is still lower than the immobilized enzyme.
Question 4:
What are the yield of conversion? What amount of compounds can we get from the procedure? Is there any improvement compared to the free fluorinase?
Answer:
Thank you for your kind comments and suggestions.
The results from the Figure 9 show that the yield of conversion for SAM is more than 98%, almost 100% when the FDAS@FHAp-NFs biocatalyst is used. Calculated from the results using HPLC analysis, about 2.093 mg of 5′-fluorodeoxy adenosine (5′-FDA) can be formed after 25 hours. However, the yield of conversion is only 80% for free enzyme. In addition, at the first two hours, the conversion rate of the immobilized enzyme was twice that of free enzyme.
Round 2
Reviewer 2 Report
The authors have nicely corrected their manuscript that deserves to be reported now.
Author Response
The answers to the comments are presented in the following.
Editor
Requirements for resubmission:
As we still found some overlap sentences (even seven continuous words in one sentence) with previous works in your manuscript which is not allowed by us. So we kindly suggest you to rewrite them and lower significantly the similarity index when you make the
revisions.
Answers:
We have revised carefully the manuscript according to your suggestions and reviewer’s comments. In this revision, we tried our best to lower significantly the similarity index and especially paid attention to some overlap sentences (even seven continuous words in one sentence). However, in the references, we think the similarity can not be avoided.
Reviewer #2:
Comments and Suggestions for Authors
The authors have nicely corrected their manuscript that deserves to be reported now.
Response:
Thank you for your review and kind comments.